# Endoplasmic Reticulum Stress Contributes to Intestinal Injury in Intrauterine Growth Restriction Newborn Piglets

**DOI:** 10.3390/ani14182677

**Published:** 2024-09-14

**Authors:** Tingting Fang, Gang Tian, Daiwen Chen, Jun He, Ping Zheng, Xiangbing Mao, Hui Yan, Bing Yu

**Affiliations:** Key Laboratory of Animal Disease-Resistance Nutrition, Animal Nutrition Institute, Sichuan Agricultural University, Chengdu 611130, China; fangtingting_cwnu@163.com (T.F.); tgang2008@126.com (G.T.); chendwz@sicau.edu.cn (D.C.); hejun8067@163.com (J.H.); zpind05@163.com (P.Z.); acatmxb2003@163.com (X.M.); yan.hui@sicau.edu.cn (H.Y.)

**Keywords:** intrauterine growth retardation, endoplasmic reticulum stress, intestine, apoptosis, barrier function

## Abstract

**Simple Summary:**

Current evidence suggests that protein synthesis dysfunction within the endoplasmic reticulum (ER) of the small intestine in animals with intrauterine growth retardation (IUGR) may be a potential factor contributing to the adverse postnatal outcomes associated with intestinal growth and development. However, little is known about whether the impaired protein-folding capacity of ER triggers the adaptive cellular response known as the unfolded protein response (UPR), as well as the subsequent signaling pathways leading to cellular apoptosis in the intestines of IUGR pigs. Therefore, this study aimed to evaluate the structural integrity of the ER and elucidate the potential signaling cascade of the UPR, which may mitigate the effects of ER stress (ERS) within the intestinal mucosa of IUGR neonates. The results showed that the mitochondrial swelling and ER dilation in the intestinal mucosa, resulting from restricted intrauterine development, occur on the neonatal day. The activation of the IRE1α and PERK pathways within the UPR, in response to ERS, was observed in the intestines of IUGR newborn piglets, leading to apoptosis in intestinal cells mediated by the transcription factor CHOP. Understanding this apoptotic mechanism underlying the structural and functional impairment in the guts of animals with IUGR is crucial for identifying potential preventive strategies.

**Abstract:**

Intrauterine growth retardation (IUGR) in piglets is associated with a high rate of morbidity and mortality after birth due to gut dysfunction, and the underlying mechanisms remain poorly understood. This study selected six pairs of IUGR newborn male piglets and normal birth weight newborn piglets (Large White × Landrace) to investigate differences in intestinal structure and digestive functions, intestinal ERS and apoptosis, intestinal barrier function, and inflammatory response. The results showed that IUGR significantly reduced the jejunal villi height (*p <* 0.05) and the ratio of villus-height-to-crypt-depth (*p* = 0.05) in neonatal piglets. Additionally, the microvilli in the jejunum of IUGR neonatal piglets were shorter than those in normal-weight piglets, and swelling of the mitochondria and expansion of the endoplasmic reticulum were observed. IUGR also significantly reduced serum glucose and lactase levels (*p* < 0.05) while significantly increasing mRNA levels of jejunal *IRE1α*, *EIF2α*, *CHOP*, *Bax*, *Caspase9*, *Mucin2*, *Claudin-1*, *Occludin*, *ZO-1*, *Bcl-2*, *IL-6*, and *IFN-γ* (*p* < 0.05), as well as GRP78 protein levels in neonatal piglets (*p* < 0.05). These findings suggest that IUGR impairs intestinal structure and barrier function in newborn piglets by enhancing intestinal inflammatory responses, activating intestinal ERS and the signaling pathways related to the unfolded protein response, thereby inducing ERS-related apoptosis.

## 1. Introduction

In the context of multiple births, intrauterine growth retardation (IUGR) is a particularly prevalent complication during the perinatal period. The pursuit of larger litter sizes has led to an increased incidence of IUGR piglets, ranging from 15–25% in the intensive pig farming industry [1]. IUGR piglets face significant challenges in the first week post-birth, with a mortality rate as high as 75%, and this vulnerability is attributed to the underdevelopment of critical systems, including the nervous, respiratory, and especially the intestinal tracts [2]. The intestine, a vital organ for nutrient absorption, energy metabolism, and immune defense, is adversely affected in IUGR piglets. These piglets exhibit distinctive features of intestinal damage, characterized by shorter and less numerous intestinal villi, as well as alterations in the balance between mitosis and apoptosis within the intestinal crypts and villi [3,4,5,6]. This condition is further exacerbated by reduced activity of brush border enzymes [7], and a decrease in the gene expression of the nutrient transporter [8]. Moreover, impairment of intestinal barrier function—encompassing physical [9], immune [10], chemical [11], and microbial barriers [12]—has been reported in IUGR infants during early postnatal life. The structural and functional damage to the intestine in IUGR pig neonates extends beyond the post-weaning period, resulting in long-term and potentially permanent adverse effects on the piglet’s adaptability to the environment, feed conversion rates, reproductive capabilities, carcass quality, and overall productivity.

Similarly, IUGR piglets exhibit a reduced capacity to maintain cellular calcium homeostasis and endoplasmic reticulum (ER) function in the gut compared to their normal-weight counterparts [13]. The ER, a vital membranous organelle in mammalian cells, plays a critical role in sustaining calcium equilibrium, cholesterol production, and lipid synthesis. In addition to impairments in calcium homeostasis, the cholesterol and lipid synthesis functions are compromised in IUGR piglets [13]. Furthermore, the ER is essential for the proper folding and post-translational modification of membrane-bound and secretory proteins [14]. In the small intestine of newborn IUGR piglets, the ER’s protein-folding function is significantly impaired, resulting in increased expression of the ER chaperone glucose regulated protein 78 (GRP78) and protein disulfide isomerase-associated 3 (PDIA3) [13,15]. This misfolding can lead to a backlog of unfolded or misfolded proteins within the ER, triggering a series of downstream reactions known as ER stress (ERS) [16]. In response, cells attempt to mitigate ERS by initiating an adaptive cellular response known as the unfolded protein response (UPR), a signaling cascade from the ER to the nucleus designed to restore the normal ER protein-folding function [14]. However, limited studies have focused on the relationship between IUGR and ERS in the intestines of newborn piglets, and the specific UPR pathways involved, as well as whether they lead to subsequent apoptosis, remain unclear.

Therefore, in the current study, a newborn IUGR and normal birth weight (NBW) pig model was used, utilized to systematically establish the structural and functional disparities between their intestines in terms of morphology, barrier function, inflammation levels, ERS, and apoptosis, thereby elucidating the potential mechanisms underlying IUGR-induced intestinal damage. The results may enhance our understanding of gut development and adaption in IUGR piglets, which holds substantial significance for improving the economic viability of the swine industry.

## 2. Materials and Methods

### 2.1. Animals and Sample Collection

All experimental procedures used in this study were performed in accordance with the guidelines of the National Research Council’s Guide for the Care and Use of Laboratory Animals, and they were approved by the Animal Care and Use Committee of Sichuan Agricultural University (China) (Case No., SYXK (Sichuan) 2019-187). A total of fifteen Large White × Landrace pregnant sows with an average gestation period of 114 ± 1 days were selected and housed in farrowing crates until parturition. The sows were fed 3 kg of feed twice daily and had free access to drinking water during the experimental period. The newborn piglets were weighed immediately after delivery, and those with an average birth weight of more than 2 standard deviations below the average birth weight of the total population were defined as IUGR piglets. Based on birth weight, six pairs of male newborn piglets—each consisting of one low birth weight (LBW) piglet, namely IUGR, and one normal birth weight (NBW) piglet—were selected from six litters. Piglets with a birth weight of 1.49 ± 0.06 kg were classified as NBW, while those weighing 0.80 ± 0.20 were identified as IUGR. After weighing, all piglets were euthanized for blood and intestine sample collection within the first 2–4 h of life.

Blood samples were collected from the anterior vena cava of all newborn piglets, centrifuged at 4 °C and 3000 rpm for 10 min, then the serum was collected and stored at −20 °C for subsequent analysis. The small intestine without the mesentery was immediately collected and allocated into the jejunum, as described by a previous study [17]. Jejunal tissue samples (2–4 cm) were fixed in 4% paraformaldehyde solution for morphological analysis. For electron microscopy, 1–2 cm sections of the jejunum were immersed in a 2.5% glutaraldehyde solution for fixation. The mucosal scrapings of the jejunum were rapidly frozen using liquid nitrogen, and subsequently stored at −80 °C for further analysis.

### 2.2. Serum Glucose Analysis

Serum glucose concentration was determined using a hexokinase colorimetric assay with a commercial glucose assay kit, and the absorbance was recorded at 340 nm. The testing procedure was carried out following the manufacturer’s instructions (Nanjing Jiancheng Bioengineering Institute, Nanjing, China).

### 2.3. Morphological Analysis

After being fixed in paraformaldehyde solution, the jejunal samples were rinsed with physiological saline, and subsequently embedded in paraffin wax. Once sectioned, the slides were deparaffinized and hydrated prior to staining with hematoxylin and eosin (H&E). Ten well-orientated and intact villi and crypts were captured and measured with an image processing and analysis system (Image-Pro Plus 6.0, Media Cybernetics, Rockville, MD, USA). The mean heights of the villi, depths of the crypts, and the critical villus-height-to-crypt-depth ratio were calculated.

### 2.4. Electron Microscope Analysis

The jejunal tissues were fixed with 2.5% glutaraldehyde solution for 24 h, and then incubated with osmium tetroxide (1%) for 1.5 h. Then, the jejunal tissues underwent dehydration at room temperature, followed by infiltration embedding and polymerization, and the specimens were cut into ultrathin sections (80 nm). The sections were then stained with uranyl acetate and lead citrate for 15 min, mounted, and examined under a transmission electron microscope for photography and observation [18].

### 2.5. Terminal Deoxynucleotidyl Transferase-Mediated Deoxyuridine Triphosphate Nick End Labeling (TUNEL) Assay

The apoptotic index of the jejunal cells was determined using the TUNEL assay. After fixation, the jejunal tissue was dehydrated, embedded, and sectioned. Paraffin sections underwent antigen retrieval, and following the quenching of endogenous peroxidase activity, the sections were treated with TdT working solution. After incubation with Streptavidin-HRP reagent, the sections were developed with DAB (3,3′-Diaminobenzidine) for coloration, counterstained with hematoxylin, and mounted for observation using an image processing and analysis system (Image Pro Plus, Media Cybernetics, Bethesda, MD, USA). The apoptotic rate was calculated as the number of TUNEL-positive cells divided by the total number of cells in each field, and the results were averaged [19].

### 2.6. Jejunal Digestive Enzyme Measurement

Total protein content and activities of trypsin, lipase, and lactase in the jejunal tissue were determined in mucosal homogenate. After thawing, the mucosal scrapings were homogenized with saline solution (1:9, weight–volume) and centrifuged at 3000× *g* for 15 min. The total protein content of the samples was determined using the Coomassie Brilliant Blue method. Trypsin, lipase, and lactase activities were measured using assay kits according to the kit instructions (Nanjing Jiancheng Bioengineering Institute, Nanjing, China).

### 2.7. RNA Extraction and Real-Time PCR Assay

A reverse transcription polymerase chain reaction (RT-PCR) was employed to determine the expression levels of inflammation-related genes, including interleukin-1β *(IL-1β*), *IL-6*, *IL-8*, *IL-10*, tumor necrosis factor-α (*TNF-α*), and interferon-γ (*IFN-γ*), as well as ERS-related genes such as *GRP78*, protein kinase RNA-activated (PKR)-like ER kinase (*PERK*), inositol requiring enzyme 1 α (*IRE1α*), eukaryotic translation initiation factor 2α (*eIF2α*), activating transcription factor 4 (*ATF4*), activating transcription factor 6 (*ATF6*), and X-box-binding protein 1 (*XBP1*)]. Additionally, intestinal barrier function-related genes (*Mucin2*, *Claudin1*, *Occludin*, *ZO-1*) and apoptosis-related genes [(CAAT/enhancer-binding protein (C/EBP) homologous protein (*CHOP*), B-cell lymphoma-2 (*Bcl-2*), Bcl-2-associated X protein (*Bax*), *Caspase3*, *Caspase8*, *Caspase9*] were analyzed in jejunal tissue. The primer sequences for these genes are detailed in Table 1. The total RNA of jejunal tissue was extracted using Trizol reagent, and its integrity was verified. The Prime-Script^®^ RT Master Mix kit (Takara, Dalian, China) was utilized for the reverse transcription of RNA into cDNA, according to the instructions of the manufacturer. Real-time PCR assays were performed using a TB Green™ Premix Ex Taq™ II kit (Takara, Dalian, China) on complementary DNA samples on a QuanStudio 6 Flex Real-Time PCR System (Applied Biosystems, Foster City, CA, USA). β-actin was chosen as the endogenous control gene for normalization, and the relative mRNA abundance of the analyzed genes was calculated using the 2^−ΔΔCT^ method [20].

### 2.8. Protein Expression Measurement with Western Blot

Jejunal proteins were extracted using a radioimmunoprecipitation (RIPA) assay buffer (Beyotime Biotechnology, Shanghai, China). Insoluble substances were removed by centrifugation of the lysates, and the protein concentration of the samples was determined using a BCA kit (Beyotime Biotechnology, Shanghai, China). Proteins from each sample were separated by SDS-PAGE electrophoresis and transferred to a PVDF membrane (Merck Millipore Ltd., Tullagreen, Ireland). Detection was performed using primary antibodies against GRP78 (Abcam, Waltham, MA, USA) and β-actin (Santa Cruz, CA, USA). After blocking with 5% bovine serum albumin in TBST, the PVDF membranes were incubated with specific primary antibodies overnight at 4 °C. Following rinses with TBST, the corresponding HRP-conjugated secondary antibodies (Santa Cruz, CA, USA) were used for membrane incubation. Signals were visualized using an Enhanced Chemiluminescence (ECL) assay (Bio-Rad, Shanghai, China), and scanning was performed with a ChemiDocTM XRS+ Imager System (Bio-Rad Laboratories, Inc.).

### 2.9. Statistical Analysis

All data were expressed as mean ± SD. Experimental observations were analyzed using a Student’s *t*-test after verifying normal distribution. Data analysis was performed using SPSS software version 22.0 (IBM Corporation, Armonk, NY, USA). *p* < 0.05 was considered as statistically significant; 0.05 ≤ *p* < 0.1 was considered as a trend.

## 3. Results

### 3.1. Jejunal Morphology

IUGR significantly affected the morphological characteristics of the jejunum in newborn piglets. It markedly reduced the height of the jejunal villi (Figure 1B; *p* < 0.05), and tended to decrease the ratio of villus-height-to-crypt-depth (*p* = 0.05). Electron microscopy observations indicated that the microvilli in the jejunal tissue of IUGR newborn piglets were shorter than those in NBW piglets, and they exhibited signs of mitochondrial swelling and dilation of the endoplasmic reticulum (Figure 1C).

### 3.2. Serum Glucose Level and Jejunal Digestive Enzyme Activities

As shown in Table 2, the serum glucose level and jejunal lactase activity in IUGR newborn piglets were significantly lower than those in NBW piglets (*p* < 0.05). In contrast, the activities of trypsin and lipase did not show significant differences (*p* > 0.05).

### 3.3. Jejunal ERS-Related Gene Expression and GRP78 Protein Level

IUGR resulted in a significant increase in the mRNA levels of *IRE1α* and *eIF2α* in the jejunum of newborn piglets (Figure 2A; *p* < 0.05). Concurrently, the protein levels of GRP78 in the jejunum of IUGR piglets were markedly increased (Figure 2B; *p* < 0.05). However, IUGR did not significantly affect the mRNA expression levels of other ERS-related genes, including *GRP78*, *PERK*, *ATF6*, *XBP1*, and *ATF4* (*p* > 0.05).

### 3.4. Jejunal Apoptosis

IUGR was associated with a significant increase in apoptotic levels within the jejunum of newborn piglets (Figure 3A; *p* < 0.05). The expression levels of apoptosis-related genes, including *CHOP* (Figure 3B), *Bax* (Figure 3D), and *Caspase9* (Figure 3G), were significantly higher in the jejunum of IUGR newborn piglets compared to those of normal-weight piglets (*p* < 0.05). Conversely, the expression level of the *Bcl-2* gene was significantly lower in IUGR piglets (Figure 3C; *p* < 0.05), while the mRNA expression levels of *Caspase3* (Figure 3E) and *Caspase8* (Figure 3F) showed no significant changes (*p* > 0.05).

### 3.5. Intestinal Barrier Gene Expression in the Jejunum

Table 3 demonstrated that the mRNA levels of intestinal barrier-related genes, specifically *Mucin2*, *Claudin1*, *Occludin*, and *ZO-1*, were substantially decreased in the jejunum of IUGR newborn piglets compared to their normal-weight counterparts (*p* < 0.05).

### 3.6. Expression Levels of Cytokine Genes in the Jejunum

As shown in Table 4, IUGR newborn piglets exhibited higher mRNA expression levels of *IL-6* and *IFN-γ* in the jejunum compared to NBW piglets (*p* < 0.05), with a trend towards increased *TNF-α* mRNA expression levels (*p* = 0.09). Conversely, the mRNA expression levels of *IL-1β*, *IL-8*, and *IL-10* in the jejunum of IUGR newborn piglets showed no significant differences compared to NBW piglets (*p* > 0.05).

## 4. Discussion

IUGR is a common perinatal complication in swine production, characterized by a high incidence and low survival rate. IUGR piglets, hindered by inadequate milk intake and incomplete organ developments, are particularly susceptible to infections, resulting in a high mortality risk during the neonatal period [2]. The gastrointestinal tract, an essential organ for nutrient digestion, absorption, metabolism, and immune defense, exhibits developmental impairments that underscore the significant negative impacts of IUGR on newborns. Previous studies have shown that IUGR piglets experience a 15%–20% reduction in the average number of villi and the villus height per unit area when compared to normal-weight piglets [21]. In this study, IUGR was found to significantly reduce the height of jejunal villi, and it exhibited a tendency to decrease the villus-to-crypt ratio in newborn piglets, indicating severe morphological damage to the jejunal villi. Subsequent scanning electron microscopy observations revealed that the microvilli of IUGR piglets were shorter, and the mitochondria displayed signs of swelling, while the endoplasmic reticulum appeared irregularly dilated. Mitochondrial swelling is associated with disruptions in oxidative phosphorylation processes, indicating a diminished antioxidant capacity in the body [22], which may contribute to oxidative stress due to a redox imbalance in the intestines of IUGR newborn piglets [23]. The endoplasmic reticulum (ER) is a membranous tubular organelle distributed throughout the cytoplasm, playing a crucial role in protein-folding, modification, and maturation [24]. IUGR can affect the initiation of protein translation in piglets, reduce post-translational modifications, weaken protein-folding capacity, and subsequently lead to ER dysfunction [17]. Therefore, the abnormal morphology of the ER in this study may serve as a significant external indicator of impaired ER function in the intestines of IUGR piglets.

Dysfunction in protein-folding can result in the accumulation of misfolded or unfolded proteins within the ER, triggering ERS and initiating a series of unfolded protein response (UPR) reactions [14]. Previous studies have indicated that the expression of the ERS-associated protein GRP78 was elevated in the intestines of IUGR newborn piglets, thereby triggering intestinal ERS [15]. GRP78 is an important molecular chaperone protein located in the ER. Under physiological conditions, GRP78 binds to three sensory proteins, IRE1α, PERK, and ATF6, keeping them in an inactive state [25,26]. However, under conditions of ERS, GRP78 dissociates from these sensors, activating a signaling cascade that prevents the aggregation of unfolded or misfolded proteins [16]. Thus, the upregulation of GRP78 is commonly regarded as an indicator of ERS activation. In this study, we observed a significant increase in GRP78 protein expression and the mRNA levels of *IRE1α* and *eIF2α* in the jejunum of IUGR newborn piglets. This indicates that IUGR not only disrupts the structure of the ER in the intestines of newborn piglets, but also induces intestinal ERS, primarily by activating the IRE1α and PERK signaling pathways. However, only GRP78 protein expression was observed in this study; as such, this may not be enough to account for the intricate gene–protein interactions and regulatory networks that are typically involved in the UPR cellular responses. Thus, additional experiments should be conducted, with a focus on the underlying mechanism for the impact of the UPR response on eliminating the intestinal ERS of IUGR.

Considering the potential toxicity associated with the accumulation of unfolded and misfolded proteins in ERS cells, persistent and excessive ERS can activate either the intrinsic or extrinsic pathways of apoptosis. Both pathways trigger the activation of caspase protease, with all branches of the UPR-IRE1α, PERK, and ATF6 playing roles in apoptosis [27]. The results of this study indicate that IUGR significantly increases the apoptotic level in the jejunum of newborn piglets and elevates the expression of the following apoptosis-related genes: *CHOP*, *Bax*, and *Caspase9*. CHOP is a transcription factor primarily initiated by the PERK–CHOP and ATF6–CHOP pathways in the UPR, and it can also be induced by IRE1α [16]. Overexpression of CHOP can lead to the induction of pro-apoptotic factors such as Bcl-2, DR5, and Bim, which, in turn, increase the expression of apoptotic molecules like Caspase, Bax, and Bak, thereby activating both the extrinsic and mitochondrial apoptosis pathways [16]. Collectively, these results demonstrate that IUGR not only induces ERS in the intestines of newborn piglets, but it also triggers CHOP-mediated intestinal ERS-induced apoptosis.

In the early stages of life, colostrum and milk serve as the primary nutritional sources for newborn infants, playing a crucial role in intestinal development, nutrient absorption, and immune protection [28]. IUGR can lead to incomplete development of intestinal morphology and increased cellular apoptosis, significantly impacting the digestive and absorptive capabilities for nutrients, alongside variations in the expression of brush border enzymes and transport proteins [29]. This study found that lactase activity in the jejunum of IUGR newborn piglets was significantly diminished, accompanied by reduced serum glucose levels, suggesting impaired digestion and utilization of bioactive substances in milk during the early neonatal stages.

After birth, the intestinal epithelial cells of newborn piglets continue to proliferate and migrate, but they are not yet fully developed, resulting in an immature intestinal epithelial barrier function. In IUGR newborn piglets, the gene expression levels of *Mucin2*, *Claudin-1*, *Occludin*, and *ZO-1* in the jejunum were lower than those of their NBW littermates in the present study. Wang et al. [23] also observed reduced expression of Claudin-1 and Occludin in the small intestine of IUGR piglets shortly after birth. Claudins, Occludin, and ZO-1 are unique markers of tight junction integrity found in the intestinal epithelial barrier [30], and their expression is closely related to the physiological state of the animal. Mucin2, a secreted mucin produced by goblet cells, serves as an important barrier that isolates intestinal tissues and the immune system from the microbiota and contents in the intestinal lumen, and it also plays a crucial role in resisting pathogens [31]. The decreased expression of tight junction proteins and Mucin2 may indicate a reduction in the integrity of the physical and chemical barriers in the intestines of IUGR newborn piglets.

In this context, we also observed increased levels of *IL-6* and *IFN-γ* in the jejunum of newborn piglets, along with a trend towards elevated gene expression levels of *TNF-α*. Cytokines are key factors that regulate the function of the intestinal mucosal barrier, affect the migration and activation of immune cells, and regulate cellular metabolism to maintain intestinal homeostasis [32]. An increase in the expression of pro-inflammatory cytokines and a decrease in the synthesis of anti-inflammatory cytokines have previously been observed in the intestines of IUGR piglets during early neonatal stages [10,11]. Moreover, Zhang et al. [33] noted that inflammatory processes are more pronounced in IUGR animals compared to their normal littermates. This peculiarity, combined with the impaired integrity of the intestinal barrier function, may constitute the early onset of enteritis in IUGR neonates [34].

Considering the interplay between intestinal functional homeostasis of the host and health, more particularly, the possible physiological influences of the IUGR-induced ERS on intestinal growth and development, the strategies to promote postnatal growth and health of the livestock should be initiated at the key stages of prenatal and postnatal development. For example, enhancing maternal nutritional status can serve as an effective strategy to promote fetal development in immature pregnant animals, and will lessen the incidence of neonatal morbidity in infants with IUGR [1]. Timely nutritional interventions aimed at neonates during the early postnatal period could alleviate the adverse intestinal health consequences and contribute to sustaining the subsequent physiology and metabolic status of IUGR [35].

## 5. Conclusions

In summary, IUGR triggers the activation of two UPR signaling pathways, IRE1α and PERK, in response to ERS in the intestines of newborn piglets. This activation leads to ERS-mediated apoptosis in the intestine, orchestrated by the molecular CHOP. Additionally, IUGR reduces the height of the intestinal villi and lactase activity, increases intestinal inflammatory responses, and compromises the structural integrity and barrier functions of the intestines in piglets. Further research focusing on ERS is needed to explore the pathogenesis of IUGR-induced intestinal damage, ultimately leading to the development of strategies aimed at ERS that promote animal growth and health through enhanced intestinal development.

## Figures and Tables

**Figure 1 animals-14-02677-f001:**
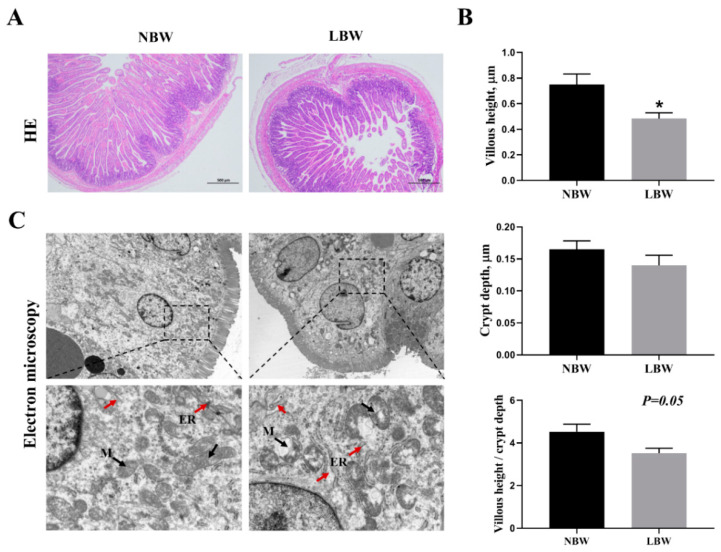
Effects of IUGR on jejunal morphology in newborn piglets. (**A**) HE staining (×40); (**B**) villus height, crypt depth, and villus height—crypt depth, *n* = 6; (**C**) scanning electron microscopy (5 µm, 1 µm), red arrow indicated endoplasmic reticulum (ER), black arrow indicated mitochondria (M). NBW: normal birth weight. LBW: low birth weight. HE: hematoxylin and eosin. * Means significant difference between two groups, *p* < 0.05.

**Figure 2 animals-14-02677-f002:**
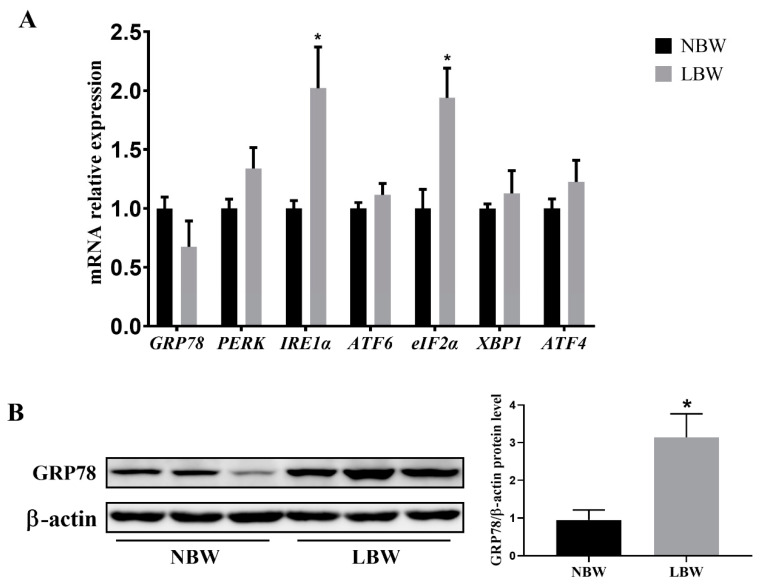
Effects of IUGR on jejunal ERS in newborn piglets. (**A**) mRNA levels of *GRP78, PERK, IRE1α*, *ATF6*, *eIF2α*, *XBP1*, and *ATF4*, n = 6; (**B**) protein abundance of GRP78, *n* = 3. NBW: normal birth weight. LBW: low birth weight. * Means significant difference between two groups, *p* < 0.05.

**Figure 3 animals-14-02677-f003:**
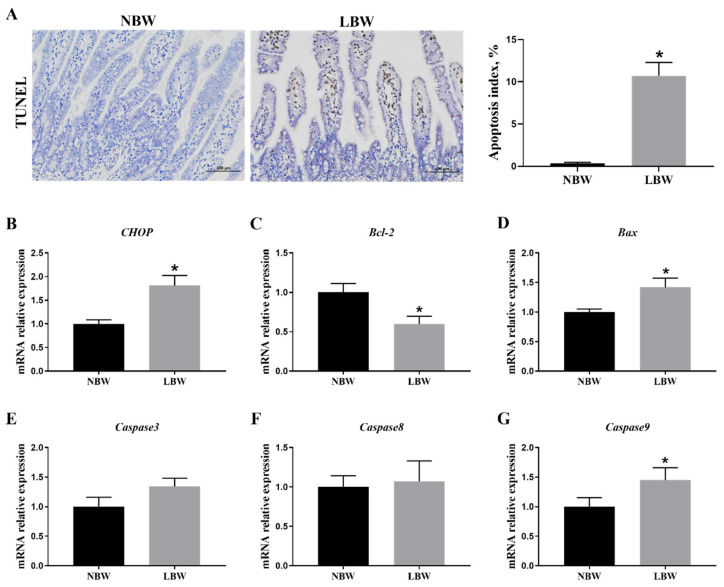
Effects of IUGR on jejunal apoptosis in newborn piglets. (**A**) Apoptosis index by TUNEL (100 µm); (**B**–**G**) mRNA levels of *CHOP*, *Bax*, *Bcl-2*, *Caspase3*, *Caspase8*, and *Caspase9*. NBW: normal birth weight. LBW: low birth weight. TUNEL: terminal deoxynucleotidyl transferase-mediated deoxyuridine triphosphate nick end labeling. * Means significant difference between two groups, *p* < 0.05, *n* = 6.

**Table 1 animals-14-02677-t001:** Sequence of primers.

Gene	Primer Sequences (5′-3′)	Size (bp)	Accession No.
*IL-1β*	F: CGTGCAATGATGACTTTGTCTGT	112	NM_214055.1
	R: AGAGCCTTCAGCATGTGTGG		
*IL-6*	F: TTCACCTCTCCGGACAAAAC	122	NM_001252429.1
	R: TCTGCCAGTACCTCCTTGCT		
*IL-8*	F: AGTTTTCCTGCTTTCTGCAGCT	144	NM_213867.1
	R: TGGCATCGAAGTTCTGCACT		
*IL-10*	F: TAATGCCGAAGGCAGAGAGT	134	NM_214041.1
	R: GGCCTTGCTCTTGTTTTCAC		
*TNF-α*	F: CGTGAAGCTGAAAGACAACCAG	121	NM_214022.1
	R: GATGGTGTGAGTGAGGAAAACG		
*IFN-λ*	F: TGCATCACATCCACGTCGAA	131	NM_001142837.1
	R: GCAGCCTTGGGACTCTTTCT		
*GRP78*	F: ACCAAAATCGCCTGACACCT	90	XM_001927795.5
	R:TGCGCTCCTTGAGCTTTTTG		
*IRE1α*	F:GAGCAGCCTTAACCCACACT	80	XM_005668695.1
	R:GTACCCGCCAGACACTCAAA		
*eIF2α*	F:GCGAAAACTAAAGATGGCGAGA	101	XM_005656337.1
	R:AGACCCGGCATTCATAGAGT		
*XBP1*	F: GCTTGGGGATGGATGCCTTA	116	NM_001142836.1
	R: CTGCAGAGGTGCACGTAGTC		
*PERK*	F:AGACTGTGACTTGGAGGACG	151	NM_001161638.1
	R:GGATGCGTTATCACAGCCAG		
*ATF4*	F:TGGCGTATTAGAGGCAGCAG	146	NM_001123078.1
	R:TTTGTCGGTTACAGCAACGC		
*ATF6*	F:CCGAAGAGAAGAGCCATCTG	127	XM_001924512.4
	R:TCCTTTGATTTGCAGGGTTC		
*Mucin2*	F: GGTCATGCTGGAGCTGGACAGT	181	XM_003122394.1
	R: TGCCTCCTCGGGGTCGTCAC		
*Claudin-1*	F: ATTTCAGGTCTGGCTATCTTAGTTGC	214	NM_001244539.1
	R: AGGGCCTTGGTGTTGGGTAA		
*Occludin*	F: AACTTCCACTGATGTCCCCCGT	138	NM_001163647.2
	R: CCTAGACTTTCCTGCTCTGCCC		
*ZO-1*	F: CGTGTCAACGCCACTATCA	90	NM_001206404.1
	R: TTGTCTTCCAAAGCCCCT		
*CHOP*	F: GTCATTGCCTTTCTCCTTCGG	139	NM_001144845.1
	R: GGTTTTTGACTCCTCCTCATTTCC		
*Bax*	F:CTGACGGCAACTTCAACTGG	200	XM_003127290.5
	R:CGTCCCAAAGTAGGAGAGGA		
*Bcl-2*	F:AGCATGCGGCCTCTATTTGA	120	XM_021099593.1
	R:GGCCCGTGGACTTCACTTAT		
*Caspase3*	F: TGTGTGCTTCTAAGCCATGG	158	NM_214131.1
	R: AGTTCTGTGCCTCGGCAG		
*Caspase8*	F: AGACAAGGGCATCATCTACGG	103	NM_001031779.2
	R: GGGTTTACCAAGAAGGGAAGG		
*Caspase9*	F: AATGCCGATTTGGCTTACGT	195	XM_003127618.4
	R:CATTTGCTTGGCAGTCAGGTT		
*β-actin*	F: GGATGACGATATTGCTGCGC	190	XM_003124280.5
	R: GATGCCTCTCTTGCTCTGGG		

**Table 2 animals-14-02677-t002:** Effects of IUGR on the serum glucose and digestive enzyme activities of jejunum in newborn piglets.

Item	NBW	LBW	*p*-Value
Glucose, mmol/L	7.02 ± 0.27	3.99 ± 0.27 *	0.00
Lipase, U/mg protein	2.40 ± 0.60	1.31 ± 0.14	0.11
Lactase, U/mg protein	63.22 ± 9.64	30.96 ± 5.12 *	0.02
Trypsin, U/mg protein	6.54 ± 1.36	7.06 ± 1.49	0.80

NBW: normal birth weight. LBW: low birth weight. * Means significant difference between two groups, *p* < 0.05, *n* = 6.

**Table 3 animals-14-02677-t003:** Effects of IUGR on the mRNA levels of jejunal barrier function in newborn piglets.

Item	NBW	LBW	*p*-Value
*Mucin2*	1.00 ± 0.13	0.58 ± 0.07 *	0.02
*Claudin1*	1.00 ± 0.14	0.40 ± 0.02 *	0.00
*Occludin*	1.00 ± 0.05	0.31 ± 0.04 *	0.00
*ZO-1*	1.00 ± 0.09	0.25 ± 0.01 *	0.00

NBW: normal birth weight. LBW: low birth weight. * Means significant difference between two groups, *p <* 0.05, *n* = 6.

**Table 4 animals-14-02677-t004:** Effects of IUGR on the mRNA levels of jejunal cytokines in newborn piglets.

Item	NBW	LBW	*p*-Value
*IL-1β*	1.00 ± 0.21	1.42 ± 0.14	0.13
*IL-6*	1.00 ± 0.20	2.30 ± 0.33 *	0.00
*IL-8*	1.00 ± 0.20	1.14 ± 0.17	0.59
*IL-10*	1.00 ± 0.19	0.65 ± 0.08	0.16
*TNF-α*	1.00 ± 0.08	1.55 ± 0.29	0.09
*IFN-γ*	1.00 ± 0.19	1.56 ± 0.12 *	0.03

NBW: normal birth weight. LBW: low birth weight. * Means significant difference between two groups, *p <* 0.05, *n* = 6.

## Data Availability

The authors declare that all data used in the research will be available and without access restrictions to those who request them.

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
