# Peer review of "Endoplasmic Reticulum Stress Contributes to Intestinal Injury in Intrauterine Growth Restriction Newborn Piglets"

_animals, 2024, doi:10.3390/ani14182677_

Round 1

Reviewer 1 Report

Comments and Suggestions for Authors

Authors report their research findings regarding IUGR newborn piglets compared to normal weight newborn piglets' difference in their intestinal structure and digestive functions, intestinal ERS and apoptosis, intestinal barrier function, and inflammatory response.

The main research aimed to evaluate the structural integrity of the ER and elucidate the potential signaling cascade of the UPR that may mitigate the effects of ER stress (ERS) within the intestinal mucosa of IUGR neonates.This topic is quite original and relevant to the field because based on the current literature most studies focus on the brain. Based on this study, authors provide clear evidence that ER stress clearly contributed to intestinal injury in IUGR piglets. This finding can help future research to prevent the impairment in IUGR gut. Most of the methodology is appreciated except PCR design needs improvement that I have mentioned in the original comments. The references, tables and figures are appropriate.

Here are the major issues and questions regarding this manuscript.

1.       Authors mentioned XBP1 gene in the section 2.7, however, no primers sequence information found in Table 1. I also strongly suggest providing PCR product size in bp for each gene in Table 1. Also, lots of typing mistakes for accession no for each gene for example (-) is missing in many accessions no.

2. The most serious problem in Table 1 is regarding the gene information does not match with the accession no provided by the authors. Here is the list of genes do not match with accession no (IL-8, IFN-gamma, GRP78, IRE1alpha, eIF2alpha, PERK, ATF4, ATF6, Claudin-1, ZO-1 and CHOP). Without the correct matching, the primer sequences provided by the authors are not the same genes and seriously affect the result in this study. 

Author Response

Comments1: Authors mentioned XBP1 gene in the section 2.7, however, no primers sequence information found in Table 1. I also strongly suggest providing PCR product size in bp for each gene in Table 1. Also, lots of typing mistakes for accession no for each gene for example (-) is missing in many accessions no.

Response1:Thank you for pointing this out. We agree with this comment. Therefore, we have added the primer sequence of XBP1 gene, PCR product size in bp for each gene, and correct accession no with (-) in Table 1.

Comments2: The most serious problem in Table 1 is regarding the gene information does not match with the accession no provided by the authors. Here is the list of genes do not match with accession no (IL-8, IFN-gamma, GRP78, IRE1alpha, eIF2alpha, PERK, ATF4, ATF6, Claudin-1, ZO-1 and CHOP). Without the correct matching, the primer sequences provided by the authors are not the same genes and seriously affect the result in this study.

Response2:  We thank the reviewer for the valuable suggestions, and we have revised the accession no to match the gene information in Table 1 section carefully in accordance with the comment.

Reviewer 2 Report

Comments and Suggestions for Authors

Intrauterine growth retardation in piglets is associated with a high rate of morbidity and mortality after birth due to gut dysfunction, and the underlying mechanisms remain poorly understood. The present study has investigated the effects of IUGR on the intestinal structure and digestive functions, intestinal ERS and apoptosis, intestinal barrier function, and inflammatory response. The topic is interesting. The following revision could improve the quality of the paper.

The paper did not labeled lines thus it is hard to write the comments.

Abstract:

More information about the piglets needs to be added, including the species, sex, age etc.

Please checking the writing of the abbreviation of the genes when express the changes of mRNA. Usually, the only the first letter should be capital.

Methods:

Please add some references for the methods. Many of them do not have specific references.

Results:

How about the body weight information for the IUGR? This is important information.

Figure 1, please add the full name for the abbreviation in the figure legends. Please check the similar issues throughout the paper.

Table 2, please correct ‘mmol/l’ to ‘mmol/L’.

The quality of presentation all the tables needs to be improved. Too much space could be reduced.

Why only one gene was analyzed their changes in the protein level? This reviewer suggested added at least 4-5 key genes changes in the protein level by WB and/or enzyme activities changes.

Dissuasion:

What is the limitation of the findings? What need to be done in the future? How could the industry use it? Such as, based on the findings, what kind of the nutrients or other strategy could be used to mitigated the negative effects of IUGR? The authors should make some explanation for it.

Author Response

Comments1: Abstract: More information about the piglets needs to be added, including the species, sex, age etc. Please checking the writing of the abbreviation of the genes when express the changes of mRNA. Usually, the only the first letter should be capital.

Response1: We thank the reviewer for the valuable suggestions, and we have revised the Abstract section carefully in accordance with the comment. 1) We have added the species, sex, and age information about the piglets used in this study in the Abstract section of our manuscript. 2) We have revised the writing of the abbreviation of the genes with first letter capital.

Comments2: Methods: Please add some references for the methods. Many of them do not have specific references.

Response2: Thank you for your suggestion, we have added some references for the methods in the manuscript.

Comments3: Results: How about the body weight information for the IUGR? This is important information.

Response3:  Thank you for your suggestion, we provided the body weight information for the IUGR in the Animals and Sample Collection section. In this study, the newborn piglets were weighed immediately after delivery, and those with an average birth weight more than 2 standard deviations below the average birth weight of the total population were defined as IUGR piglets. Based on birth weight, six pairs of male newborn piglets - each consisting of one low birth weight (LBW) piglet namely IUGR and one normal birth weight (NBW) piglet - were selected from six litters. Piglets with a birth weight of 1.49±0.06 kg were classified as NBW, while those weighing 0.80±0.20 were identified as IUGR. After weighing, all piglets were euthanized for blood and intestine samples collection within the first 2-4 hours of life.

Comments4: Figure 1, please add the full name for the abbreviation in the figure legends. Please check the similar issues throughout the paper.

Response4: Thank you for your suggestion, we have added the full name for the abbreviation in the figure legends of all Figures and Tables in the manuscript.

Comments5: Table 2, please correct ‘mmol/l’ to ‘mmol/L’.

Response5: Thank you for your suggestion, we have corrected the ‘mmol/l’ to ‘mmol/L’ in the Table 2 of the manuscript.

Comments6: The quality of presentation all the tables needs to be improved. Too much space could be reduced.

Response6: Thank you for your suggestion, we have reduced the superfluous space in all the tables to improve the quality of presentation in the manuscript.

Comments7: Why only one gene was analyzed their changes in the protein level? This reviewer suggested added at least 4-5 key genes changes in the protein level by WB and/or enzyme activities changes.

Response7: Thank you very much for pointing out this important issue. We agree with your comments that added at least 4-5 key genes changes in the protein level by WB and/or enzyme activities changes would be useful to further support our findings. Your suggestion provides a direction for our next research. Unfortunately, due to the limited time and resources, we are unable to conduct additional experiments to include more proteins in our analysis. Because the experiment was conducted two years ago, we cannot collect the same data samples. We understand your point that it could further strengthen our findings. However, we would like to emphasize that even without this particular experiment, our manuscript provides a comprehensive and conclusive discussion on the research question, and we will add this deficiency to the limitation section in discussion to provide a more comprehensive understanding of our work. Moreover, we plan to incorporate your thoughtful and constructive comments into our future research endeavors, as they align closely with our long-term research interests and goals. We are confident that these future studies will not only address the points you've raised but also contribute meaningfully to the broader scientific community.

Comments8: Dissuasion: What is the limitation of the findings? What need to be done in the future? How could the industry use it? Such as, based on the findings, what kind of the nutrients or other strategy could be used to mitigated the negative effects of IUGR? The authors should make some explanation for it.

Response8: In this study, the significant limitation is the analysis of only one gene and its corresponding protein level that may be insufficient to account for the intricate gene-protein interactions and regulatory networks that are typically involved in the UPR cellular responses. This may not capture the complexity of the biological processes under investigation, as multiple proteins often interact and contribute to the observed phenotypes. Thus, additional experiments will be conducted in the future to strengthen its scientific validity of our findings with a focus on the underlying mechanism for the impact of UPR response on eliminating the intestinal ERS of IUGR. Additionally, based on our findings, some strategies can be used to promote postnatal growth and health of livestock at the key stages of prenatal and postnatal development. For example, enhancing maternal nutritional status can serve as an effective strategy to promote fetal development in immature pregnant animals and will lessen the incidence of neonatal morbidity in infants with IUGR. Timely nutritional interventions aim at neonate during the early postnatal peri-od could alleviate the adverse intestinal health consequences and contribute to sustain the subsequent physiology and the metabolic status of IUGR. We have added the limitation of this findings and the strategy can be used to mitigated the negative effects of IUGR based on our findings in discussion section to provide a more comprehensive understanding of our work.

Round 2

Reviewer 1 Report

Comments and Suggestions for Authors

I have checked the revised version of the MS. The authors revised all the mistakes in the manuscript and should be accept for publication.